# EPA Is Cardioprotective in Male Rats Subjected to Sepsis, but ALA is Not Beneficial

**DOI:** 10.3390/antiox9050371

**Published:** 2020-04-29

**Authors:** Thibault Leger, Chrystèle Jouve, Isabelle Hininger-Favier, Jean-Paul Rigaudiere, Frédéric Capel, Vincent Sapin, Clarisse Moreau, Alice Charrier, Luc Demaison

**Affiliations:** 1UNH, Unité de Nutrition Humaine, UMR 1019, Université Clermont Auvergne, INRAE, CRNH Auvergne, 63000 Clermont-Ferrand, France; thibault.leger@inrae.fr (T.L.); chrystele.jouve@inrae.fr (C.J.); jean-paul.rigaudiere@inrae.fr (J.-P.R.); frederic.capel@inrae.fr (F.C.); clarisse.moreau86@yahoo.fr (C.M.); alicecharrier@hotmail.com (A.C.); 2LBFA U1055, Université Grenoble Alpes, INSERM, 38058 Grenoble, France; isabelle.hininger@univ-grenoble-alpes.fr; 3Department of Medical Biochemistry and Molecular Biology, CHU Clermont-Ferrand, 63000 Clermont-Ferrand, France; vincent.sapin@uca.fr

**Keywords:** sepsis, heart, n-3 polyunsaturated fatty acids, inflammation, oxidative stress

## Abstract

It has been proven that dietary eicosapentaenoic acid (C20:5 n-3 or EPA) protects the heart against the deleterious effects of sepsis in female rats. We do not know if this is the case for male rodents. In this case, the efficiency of other n-3 polyunsaturated fatty acids (PUFAs) remains to be determined in both female and male rats. This study aimed at (i) determining whether dietary EPA is cardioprotective in septic male rats; (ii) evaluating the influence of dietary α-linolenic (C18:3 n-3 or ALA) on cardiac function during this pathology; and (iii) finding out the physiological and molecular mechanisms responsible for the observed effects. Sixty male rats were divided into three dietary groups. The animals were fed a diet deficient in n-3 PUFAs (DEF group), a diet enriched with ALA (ALA group) or a diet fortified with EPA (EPA group) for 6 weeks. Thereafter, each group was subdivided into 2 subgroups, one being subjected to cecal ligation and puncture (CLP) and the other undergoing a fictive surgery. Cardiac function was determined in vivo and ex vivo. Several parameters related to the inflammation process and oxidative stress were determined. Finally, the fatty acid compositions of circulating lipids and cardiac phospholipids were evaluated. The results of the ex vivo situation indicated that sepsis triggered cardiac damage in the DEF group. Conversely, the ex vivo data indicated that dietary ALA and EPA were cardioprotective by resolving the inflammation process and decreasing the oxidative stress. However, the measurements of the cardiac function in the in vivo situation modulated these conclusions. Indeed, in the in vivo situation, sepsis deteriorated cardiac mechanical activity in the ALA group. This was suspected to be due to a restricted coronary flow which was related to a lack of cyclooxygenase substrates in membrane phospholipids. Finally, only EPA proved to be beneficial in sepsis. Its action necessitates both resolution of inflammation and increased coronary perfusion. In that sense, dietary ALA, which does not allow the accumulation of vasodilator precursors in membrane lipids, cannot be protective during the pathology.

## 1. Introduction

Sepsis due to microorganism infection is defined as a state of general systemic inflammation which can degenerate into septic shock, possibly leading to death. In Western societies with high financial incomes, incidences of sepsis are constantly increasing, but lethality is regressing probably due to the improvements in health care in intensive care units [1]. However, enhanced survival can be accompanied by comorbidity. Analysis of billings from 2013 revealed that sepsis accounts for nearly $24 billion in annual costs, making it the most expensive condition to treat in the entire U.S. healthcare system [2].

Systemic hypotension is one of the hallmarks of sepsis and partly results from cyclooxygenase 2 (COX2) overexpression and eicosanoid release [3]. Crucial treatment after antibiotic therapy is fluid administration to restore physiological blood pressure [4]. Septic shock is defined as a sepsis resistant to fluid resuscitation [5]. It is characterized by a mortality rate ranging from 20 to 45% and nearly 50% of the deaths occur as a consequence of heart failure [6,7].

Sepsis-related depression of myocardial function is due to bacterial lipopolysaccharides [8]. Endotoxins bind to cardiac Toll-like receptors (TLRs) and activate the NF-κB pathway, leading to pro-inflammatory cytokine overexpression [9]. These last (tumor necrosis factor alpha (TNF-α), interleukin-1β (IL-1β), interleukin-6 (IL-6), etc.) belong to the immune response system and lead to the destruction of the microorganism. However, they are also able to damage host cells and to trigger heart failure [10].

Polyunsaturated fatty acids (PUFAs) of the n-3 families have been recognized as cardioprotective in several pathological situations including arrhythmias [11], myocardial stunning [12] and hypertrophy [13]. Regrettably, the diet in Western societies is too rich in n-6 PUFAs and does not contain enough n-3 PUFAs [14]. Yet, numerous studies report the beneficial effects of n-3 PUFAs during sepsis [15,16,17] due to the n-3 PUFA-related anti-inflammatory properties which contrast with the pro-inflammatory actions of n-6 PUFAs [18]. In a recent study performed in female rats subjected to cecal ligation and puncture (CLP), we partially highlighted the mechanism of the eicosapentaenoic acid (C20:5 n-3 or EPA)-related cardioprotective effect [19]. It is triggered by an improved oxygen metabolism preconditioning the heart against the deleterious effects of excessive inflammation. However, females may also be protected by their hormonal status. Dietary EPA is protective during sepsis, but we do not know whether other n-3 PUFAs such as α-linolenic (C18:3 n-3 or ALA) acid is efficient. Indeed, in contrast with dietary EPA, ALA does not allow the easy accumulation of EPA in membrane phospholipids [20]. Phospholipid EPA is directly accessible to the cyclooxygenase enzyme via the action of phospholipase A2. This leads to the formation of vasoactive eicosanoids which can increase the coronary flow [21]. Dietary ALA though favors the accumulation of docosahexaenoic acid (C22:6 n-3 or DHA) which blocks the cyclooxygenase enzyme [22], thus inhibiting the formation of vasoactive compounds and reducing the coronary flow. The heart acts like an engine. Reducing the coronary flow decreases cardiac contraction resulting in depressed cardiac function despite normal cardiomyocyte health. Therefore, dietary ALA could protect the cardiomyocytes while reducing myocardial contractility.

This study set out to (i) determine whether dietary EPA is cardioprotective in male rats made septic by cecal ligation and puncture (CLP), (ii) estimate the influence of dietary ALA in septic male rats, (iii) and evaluate the physiological and molecular mechanisms underlying the effects of these PUFAs. For this purpose, three groups of male Wistar rats were fed a diet each differing in their PUFA composition: the first one mimicking the Western diet was rich in n-6 PUFAs and deficient in n-3 PUFAs, the second one was fortified with EPA and the third one was supplemented with ALA. After 6 weeks of feeding, sepsis was induced by CLP and, exactly 24 h later, cardiac function was evaluated in in vivo and ex vivo. The data were enhanced by the measurement of several of the parameters involved in the inflammation process and production of oxidative stress as well as those describing the fatty acid profile of circulating and membrane lipids. 

## 2. Materials and Methods

### 2.1. Ethical Approval

All experiments followed the European Union recommendations concerning the care and use of laboratory animals for experimental and scientific purposes. All animal work was approved by the local board of ethics for animal experimentation (Comité d’éthique pour l’expérimentation animale Auvergne) and notified to the research animal facility of our laboratory (authorization n°APAFIS#2213-2016082409264678 v2).

### 2.2. Experimental Animals and Diet

Sixty 4 month-old male Wistar rats (Janvier, Le Genest Saint Isle, France) were maintained 3–4 per cage under controlled lightening, hygrometry and temperature conditions. Each rat received a n-3 PUFAs-deficient diet for six weeks. After this period, the animals were randomly allocated to three dietary groups for 6 weeks. The three diets differed in their PUFA composition. They all contained 5% of lipids and approximately 2.25% of linoleic acid (C18:2 n-6 or LA). In each of the different diets, 0.625% of their monounsaturated fatty acid (MUFAs) were substituted with a specific PUFA: (i) LA for the n-3 PUFA-deficient diet (DEF); (ii) ALA for the ALA-rich diet; and (iii) EPA for the EPA-rich diet. The fatty acid composition of the 3 diets is presented in Table 1. The diets with n-3 PUFAs displayed a n-6/n-3 PUFA ratio close to 3. At the end of this period, half of the rats from each group was subjected to CLP to induce severe sepsis (Sept) and the other half were sham-operated (Sham). Thus, the study dealt with six sub-groups according to the diet and surgery: (i) DEF-Sham; (ii) DEF-Sept; (iii) ALA-Sham; (iv) ALA-Sept; (v) EPA-Sham; and (vi) EPA-Sept.

### 2.3. Surgical Procedure

CLP was performed according to Toscano et al. [23]. Briefly, the animals were anesthetized with isoflurane (induction 4%, maintenance 2%). After shaving the fur, the external abdominal wall was disinfected with alcoholic betadine and a horizontal incision was performed in the ventral wall at the level of the caecum in order to reach the abdominal cavity. The caecum was externalized and placed onto the disinfected external abdominal wall. A ligation of the caecum was carried out at 2 cm from the apex of the organ. Afterwards, two perforations were performed in the caecum lining at 1 cm interval on the upper face of the organ with a 20-gauge needle and soft pressure was applied to the caecum to facilitate externalization of digestive matter to the outer surface of the organ. The caecum was reinserted immediately after into the abdominal cavity and buprenorphine (0.05 mg/kg of body weight) was injected subcutaneously into the neck. Thereafter, the peritoneum was closed with 6.0 silk sutures and the skin with sutures and metal clips. Sham-operated animals were treated identically as compared to septic animals except that the caecum was not subjected to ligation and puncture. Following the operation, anesthesia was stopped and the rats were placed one per cage in the animal facility where the waking occurred between 3 and 5 min after isoflurane arrest. Each rat was then kept in a separate cage for 24 h. One rat died during anesthesia and two septic animals did not survive the 24 h post-surgery period.

### 2.4. Evaluation of Body Composition

The animals body weight along with the body composition were determined by nuclear magnetic resonance imaging with an adequate spectrometer (Echo MRI LL, Houston, TX, USA) just before the surgical procedure and at the end of the 24 h postoperative period. The data thus enabled the measurement of the lean body mass, fat mass and total body water in the animals. After the postoperative period, the weights of abdominal (perirenal plus visceral adipose tissues) and epididymal fats were measured.

### 2.5. Appraisal of In Vivo Cardiac Function

Twenty-four hours after the CLP, body composition was assessed and the in vivo cardiac function was appraised using a Millar pressure probe (Harvard Apparatus, Les Ulis, France) which was introduced into the aorta and left ventricle cavity. After anesthesia with ketamine (100 mg/kg of body weight) and xylazine (20 mg/kg of body weight), heparinization of the rats was carried out via the saphenous vein (500 UI/kg of body weight). Then, the probe was introduced into the right carotid artery and immediately inserted into the cavity of the left ventricle. In order to ensure the stability of the cardiac function, the measurements were carried out for 10 min. They allowed the measurement of the left ventricular developed pressure (LVDP), contraction and relaxation (dP/dt max and dP/dt min, respectively) and heart rate. The systolic, diastolic and mean aortic pressure was then determined by pulling out the probe from the left ventricle into the aorta. Then, the probe was removed and the right carotid was bound. Blood was collected from the abdominal aorta and centrifuged (1800 g, 10 min, 4 °C) for plasma preparation. Once obtained, the plasma was immediately frozen at the temperature of liquid nitrogen and stored at −80 °C to await the performance of biochemical analyses.

### 2.6. Assessment of Ex Vivo Cardiac Function

Directly after the evaluation of the in vivo cardiac function (24 h after the surgery), the ex vivo perfusion was performed in standardized conditions according to the non-recirculating Langendorff method. A rapid thoracotomy was carried out and the heart was immediately collected and placed in 4 °C saline buffer until cessation of heartbeat. It was then immediately (in the first minute after the removal of the heart in order to avoid cellular damages and preconditioning) perfused by the aorta at 37 °C with a Krebs-Henseleit buffer composed of (in mM) NaCl (119), KCl (4.8), MgSO_4_ (1.6), NaHCO_3_ (22), KH_2_PO_4_ (1.2), CaCl_2_ (1.8), D-glucose (11) and sodium hexanoate (0.5), pH 7.4. The perfusion buffer was maintained at 37 °C with a water-bath and constantly oxygenated with carbogen (95% O_2_, 5% CO_2_). Afterwards, the pulmonary artery was cannulated for the collection of coronary effluents. The heart was constantly perfused with a peristaltic pump (Gilson, Middleton, WI, USA) to keep the perfusion flow constant at 12 mL/min until the end of perfusion. From the fifth minute of perfusion, the heart was electrically paced at a rate of 370 beats/min. To evaluate heart contraction, a latex balloon attached to a pressure probe and an amplifier was inserted into the left ventricle. It was inflated until the diastolic pressure reached 10 mmHg. This system allowed the assessment of the systolic, diastolic and developed pressure as well as the heart rate; contraction (dP/dt max) and relaxation (dP/dt min). The evaluation of these two last parameters was possible by the use of perfectly standardized perfusion conditions. The perfusion pressure was also evaluated with a pressure probe inserted just before the aortic cannula. In conditions of perfusion at fixed coronary flow, the perfusion pressure corresponds to the coronary pressure. Changes in the coronary pressure reveal modifications of coronary volume (coronary dilatation, constriction and/or obliteration of coronary vessels). All the parameters of cardiac function were measured at the 30th minute of perfusion and the data were recorded and analyzed with HSE software (Hugo Sachs Elektronik, March-Hugstetten, Germany). Just before the end of the recording, samples of arterial and venous perfusion fluids were collected anaerobically for determination of oxygen partial pressures with a blood gas analyzer (Radiometer, Neuilly-Plaisance, France). Cardiac oxygen consumption was calculated by subtracting the venous oxygen partial pressure to the arterial one and by multiplying the difference by the coronary flow. The cardiac metabolic efficiency was also calculated (ratio between the rate pressure product and oxygen consumption). Finally, the heart was freeze-clamped at −196 °C and stored at −80 °C for subsequent analyses.

### 2.7. Lipid Profile

Fatty acid composition of plasma lipids and cardiac phospholipids. The measurements were performed according to a previously described method (Demaison et al., 1994). The total lipids were extracted according to Folch et al. (1957) and the phospholipids were separated from non-phosphorus lipids using a Sep-Pak^®^ Cartridge. After transmethylation with boron trifluoride methanol, the fatty acid methyl esters were separated and analyzed by gas chromatography. The determination of the fatty acid profile of the plasma lipids was performed similarly except that the phospholipids separation step was omitted.

Plasma triglycerides. The concentration of plasma triglycerides was determined by a colorimetric method. Triglycerides were hydrolyzed by lipase to free fatty acids and glycerol. The glycerol phosphorylation produced hydrogen peroxides (H_2_O_2_) which reacted with the reagents of the commercially available kit (Thermo Fisher Scientific, Asnières-sur-Seine, France) to generate quinone-imine, a colored compound measurable at 510 nm.

### 2.8. Fluid Biochemistry

The inflammatory cytokine tumor necrosis factor- α (TNF-α) was analyzed in the plasma using a commercially available enzyme-linked immunosorbent assay (ELISA) kit (Abcam, Paris, France). Lactate and pyruvate were assessed in the venous perfusion fluid according to Bergmeyer [24].

### 2.9. Gene Expression

Gene expression of TNF-α, interleukin-1β (IL-1β), interleukin-6 (IL-6), mitochondrial superoxide dismutase 2 (SOD2) and sirtuin 3 (SIRT3) proteins was measured in the cardiac homogenates. RNA extraction was performed using TRIzol^®^ (Thermo Scientific, Waltham, MA, USA) according to the manufacturer’s instructions. Chloroform was added (0.2 mL/mL of TRIzol^®^), and samples were mixed and centrifuged (12,000 g, 15 min, 4 °C). The aqueous phase containing total RNAs was collected, mixed with isopropanol to precipitate RNAs and centrifuged (12,000 g, 15 min, 4 °C). The resulting pellet was washed with ethanol 70% (*v/v*), dried and suspended in water. RNA quantification and integrity were controlled by measuring the ratio of optical density at 260 and 280 nm and agarose gel migration. For each sample, 2 µg of total RNAs were used to perform reverse transcription. The resulting products were used for reverse transcription quantitative polymerase chain reaction (RT-qPCR) to appraise gene expression. Gene expression was determined using specific primers (Table 2) and Rotor-Gene SYBR Green PCR master mix on a Rotor-Gene Q System (Qiagen, Courtaboeuf, France). Finally, mRNA quantification was calculated using the ddCT method. β-actin was used as the housekeeping gene due to its intergroup stability.

### 2.10. Western Blot Analysis

Tissues were ground three times in a mini bead beater in the presence of a lysis buffer constituted of 4-(2-hydroxyethyl)-1-piperazine ethane sulfonic acid (HEPES) 50 mM, sodium chloride 150 mM, ethylene diamine tetraacetic acid (EDTA) 10 mM, anhydrous sodium tetrabasic pyrophosphate 10 mM, β-glycerophosphate 25 mM, sodium fluoride 100 mM and anhydrous glycerol 1.086 M supplemented with phosphatase inhibitors (Sigma Aldrich, Saint-Quentin-Fallavier, France). Successive centrifugations were performed and the supernatants collected. Protein was quantified using a bicinchoninic acid assay kit (Thermo Fisher Scientific, Asnières-sur-Seine, France). For protein immunoblotting, 25 µg of proteins were loaded for separation by sodium dodecyl sulfate polyacrylamide gel electrophoresis (SDS-PAGE) and transferred onto polyvinylidene fluoride (PVDF) membranes. The membranes were then immunoblotted with the appropriate antibody to detect acetylated-superoxide dismutase 2 (Ac-SOD2, 24 kDa, 1:1000, Abcam #ab137037), nuclear factor of kappa light polypeptide gene enhancer in B-cell inhibitor alpha (IκBα, 39 kDa, 1:1000, Cell Signaling #9242), peroxisome proliferator-activated receptor gamma coactivator 1-alpha (PGC-1α, 90 kDa, 1:500, Santa Cruz sc-13067), superoxide dismutase 2 (SOD2, 22 kDa, 1:1000, Cell Signaling #13194), uncoupling protein-3 (UCP3, 34 kDa, 1:1000, Abcam #ab10985), and voltage-dependent anion-selective channel (VDAC, 32 kDa, 1:1000, Cell Signaling #4866). Antibody binding was detected using horse raddish peroxidase (HRP)-conjugated secondary antibodies and the ECL Western Blotting Substrate (Thermo Fisher Scientific, Asnières-sur-Seine, France). 

Immunoblots were visualized using a chemiluminescence imaging system (MF ChemiBIS, DNR bio-imaging systems, Jerusalem, Israel) and quantified using Multi Gauge V3.2 software. The assessments were performed on myocardial tissue. Myocardial proteins were referred to densitometry after protein coloration with Red Ponceau S stain for its intergroup stability.

### 2.11. Oxidative Stress Status

Several markers were used to determine the oxidative stress status in the plasma and heart tissue samples: (i) protein thiol residues whose disappearance reflects an increased oxidative stress were determined according to Faure and Lafond [25]; (ii) the antioxidant status was evaluated according to a global marker of the antioxidant power (Ferric Reducing Antioxidant Power, FRAP); (iii) the activity of glutathione peroxidase, a selenoenzyme involved in the protection against H_2_O_2_, was measured by the modified procedure of Gunzler [26] using tert-butyl hydroperoxide solution as a substrate instead of hydrogen peroxide; (iv) glutathione levels in the heart were evaluated (total, GSH and GSSG) using a one-step fluorimetric method with a commercially available kit (Abcam, Paris, France).

### 2.12. Statistical Analysis

Results are presented as mean ± SEM. The data were subjected to a 2-way analysis of variance describing the effects of the diet, those of sepsis and the cross-interaction between these two factors. When it was necessary, the means were compared using Duncan’s test. A probability lower than 0.05 was considered significant. All the calculations were performed using the NCSS (Number Cruncher Statistical System, 2010) software (NCSS, LLC, Kaysville, UT).

## 3. Results

### 3.1. Morphological Data

The different diets did not modify the weight of the animals, but did alter the body composition (Table 3). Indeed, the diets enriched with n-3 PUFA increased the fat mass and reduced the lean mass. The increase in the fat mass observed in the animals supplemented with n-3 PUFAs was mostly related to the peri-renal and visceral fats, but not the epididymal fat.

Sepsis did not significantly change the fat and lean body masses regardless of the dietary conditions. However, the sham-operation and CLP always triggered a loss of lean and fat masses compared to the values measured before the surgery. Interestingly, the loss of fat mass was higher with the sham- operation (−0.73 compared to −0.19 g/100 g of body weight in the CLP group, *p* = 0.003) whereas the loss of lean mass was higher with the CLP (−3.12 vs. −2.10 g/100 g of body weight in the sham-operated groups, *p* = 0.004). No significant cross-interactions between the effects of sepsis and those of dietary PUFAs on the loss of lean and fat masses were detected. The weights of the abdominal and epididymal fats were not modified by sepsis. 

Finally, heart weight was not affected by the diets or the surgical procedure (data not shown).

### 3.2. Fatty Acid Composition of Total Plasma Lipids

Feeding the sham-operated animals with the different diets did not modify the proportions of saturated (SFAs) and monounsaturated (MUFAs) fatty acids in the plasma lipids (Table 4). Conversely, the proportion of n-6 PUFAs was decreased by the ALA (−8%) and EPA (−13%) diets compared to the DEF diet. Except for the C18:2 n-6, C20:2 n-6 and C20:3 n-6 which were increased, all the other individual n-6 PUFAs were decreased by the n-3 PUFA-rich diets. The changes were more obvious for the EPA diet compared to the ALA diet. By contrast, the proportions of n-3 PUFAs were increased by the n-3 PUFA-rich diets (+459% and +775% for the ALA and EPA diets compared to the DEF diet). This was particularly strong for the C18:3 n-3, C20:5 n-3, C22:5 n-3 and C22:6 n-3 in the ALA group (+∞, +943, +563 and +361% compared to the DEF group) and C20:5 n-3, C22:5 n-3 and C22:6 n-3 in the EPA group (+5300, +1388 and +384%). All these modifications contributed to reduce the n-6 to n-3 PUFA ratio of plasma lipids (−84 and −90% for the ALA and EPA diets compared to the DEF diet).

Sepsis also induced changes in the fatty acid profile of plasma lipids. It increased the proportions of SFAs, namely C16:0 (+12, +4 and +8% in the DEF, ALA and EPA groups, *p* < 0.001, NS, *p* = 0.029 compared to the sham-operated groups) and those of MUFAs. By contrast, it strongly reduced the amount of n-6 PUFAs (−11, −9 and −11% for the DEF, ALA and EPA groups). The main fatty acids concerned were the C18:3 n-6 (−44, −36 and −20% for the DEF, ALA and EPA groups) and C20:4 n-6 (arachidonic acid (ARA), −20, −19 and −22% for the DEF, ALA and EPA groups). These decreases in n-6 PUFAs were not offset by augmented proportions of n-3 PUFAs. The n-6 to n-3 PUFA ratio was thus significantly reduced in the DEF group (−18%), but not in the ALA and EPA groups (−5 and −11%, not significant).

Two significant cross-interactions between the effects of the diets and those of sepsis were noticed. Firstly, a sepsis-induced increase in the C18:3 n-3 was observed in the ALA group, but not in the two other groups in which the C18:3 n-3 proportion was almost null. Secondly, the n-6 to n-3 PUFA ratio was significantly reduced by sepsis in the DEF group, but not in the ALA and EPA groups.

### 3.3. Plasma Triglycerides

The plasma concentrations of triglycerides were similar in the different dietary groups after the sham-operation (Figure 1). Sepsis increased their concentrations (+82, +50 and +77% in the DEF, ALA and EPA groups). No differences were observed between the 3 dietary groups.

### 3.4. In Vivo Cardiac Function

The cardiac function was estimated in anesthetized animals with a pressure gauge introduced into the aorta and left ventricular cavity via the right carotid. In sham-operated animals, the changes in dietary PUFA did not alter the systolic, diastolic and mean aortic pressure (Table 5). No changes in the parameters of the left ventricle contraction and relaxation were also observed. The LVDP, heart rate, contraction and relaxation were unchanged by the dietary lipids.

Sepsis tended to decrease the aortic pressures, but this was not significant. In contrast, our observations suggested that it stimulated the cardiac function. It increased the heart rate (+11%, *p* = 0.004), but this was significant only in the EPA group (+24 vs. +10 and −1% in the DEF and ALA groups). It also tended to increase the contraction in the DEF and EPA groups (+6 and +10%, respectively), but it reduced this parameter in the ALA group (−24%, *p* = 0.027). The relaxation was unchanged by sepsis in the DEF and EPA groups, but tended to be deteriorated in the ALA group (−20% *p* = 0.12).

### 3.5. Plasma Oxidative Stress in the In Vivo Situation

The changes in cardiac function were not explained by the modifications of the oxidative stress parameters in the plasma (Figure 2). The amounts of thiol residues, inversely proportional to the oxidative stress, were unaffected by the dietary PUFA and sepsis. Conversely, a significant cross-interaction was observed: although the amount of thiol residues were not modified by sepsis in the DEF and ALA groups, it was increased in the EPA group (+10%). Lipid peroxidation was evaluated by measuring thiobarbituric acid reactive substances (TBARS) levels and was not affected by the dietary PUFAs nor the sepsis. No significant cross-interaction was observed with the 2-way analysis of variance. However, when the effect of sepsis was evaluated using a 1-way analysis of variance in the ALA group only, a decrease in the TBARS level was noticed (−12%, *p* < 0.006). The anti-oxidative defenses (FRAP) were not affected by the treatments, but glutathione peroxidase activity was reduced by sepsis, especially in the DEF and EPA groups (−17 and −12%, respectively).

### 3.6. Fatty Acid Profile of Cardiac Phospholipids

The different diets altered the proportions of several fatty acids in the cardiac phospholipids (Table 6). For the SFAs, C18:0 was reduced by the diets enriched with n-3 PUFAs (−3 and −6% for the ALA and EPA diets compared to the DEF diet). Conversely, the dimethylacetal form of the C16:0 was increased by the n-3 PUFA-rich diets (+25 and +19% for the ALA and EPA diets compared to the DEF diet). The total proportion of MUFAs was reduced by the intake of n-3 PUFAs (−14 and −12% for the ALA and EPA diets). This was mainly observed for the C18:1 n-9 which was decreased by 20 and 19% by the ALA and EPA diets, respectively. As was expected, the total proportion of n-6 PUFAs was reduced by dietary ALA (−19%) and EPA (−24%). The decrease concerned the C20:4 n-6 (−14 and −26% for the ALA and EPA diets), C22:4 n-6 (−60 and −82%) and C22:5 n-6 (−92 and −97%). Conversely, the C18:2 n-6 (+16 and +21% for the ALA and EPA diets compared to the DEF diet) and C20:3 n-6 (+58 and +74%) were augmented by the n-3 PUFA-rich diets. The reduction of n-6 PUFA was offset by an increase in membrane n-3 PUFAs (+745 and +930% for the ALA and EPA groups). In the ALA group, the C18:3 n-3, C22:5 n-3 and C22:6 n-3 were augmented (+∞, +1508 and +627% compared to the DEF diet) and in the EPA group, the C20:5 n-3, C22:5 n-3 and C22:6 n-3 were increased (+∞, +3375 and +627%).

In general, sepsis induced an increase in the proportion of C22:5 n-6, but reduced the n-6 to n-3 PUFA ratio in the cardiac phospholipids. Interestingly, significant cross-interactions between the effects of the diets and those of sepsis were noticed for several parameters of the fatty acid profile of the cardiac phospholipids. Cross-interactions were observed for the C20:4 n-6, C22:5 n-6, total n-3 PUFAs and ratio between n-6 and n-3 PUFAs. The proportion of C20:4 n-6 was increased by sepsis in the EPA group (+7%), but not in the other dietary groups. The proportion of C22:5 n-6 was augmented in the DEF group (+21%), but not in the ALA and EPA groups. The proportion of total n-3 PUFAs was reduced by sepsis in both the ALA and EPA groups (−8 and −4%), but not in the DEF group. Finally, the n-6 to n-3 PUFA ratio was not changed by sepsis in the ALA and EPA groups, but was reduced by the pathological event in the DEF group (−23%).

### 3.7. Ex Vivo Cardiac Function

The cardiac function was measured in the ex vivo situation under standard conditions (Figure 3) when the hearts were paced at 370 beats/min. The LVDP, contraction, perfusion pressure, oxygen consumption and metabolic efficiency were unaffected by the diets in the sham-operated animals. Conversely, the relaxation rate was reduced by the diet enriched with ALA compared to that deficient in n-3 PUFAs (−21%). This was not the case for that measured with the diet enriched with EPA whose value ranged between those of the DEF and ALA groups.

Sepsis significantly decreased the LVDP in the DEF group (−21%). This was due to reductions in the rates of contraction and relaxation. By contrast, in the ALA and EPA groups, the LVDP as well as the rates of contraction and relaxation tended to be improved by sepsis. The sepsis-induced decrease in the LVDP in the DEF group was associated with a reduction in the perfusion pressure, whereas the upholding of the LVDP in the ALA and EPA groups was associated with maintained perfusion pressures.

### 3.8. Appraisal of the Mitochondrial Function

Lactate release in the venous effluent, pyruvate washout and glycolytic rate (sum of lactate plus pyruvate releases) were not affected by the diets and sepsis (data not shown). Conversely, the lactate release to oxygen consumption ratio, reflecting the part of anaerobic glycolysis in energy production, was increased by sepsis in the DEF group (+57%, *p* < 0.05) without being significantly altered in the other two groups (Figure 4A). This suggests sepsis-induced alterations of mitochondrial function in the DEF group. This was confirmed by VDAC expression (Figure 4D), which was only reduced by sepsis (−29%, *p* < 0.003 by 1-way analysis of variance) in the DEF group. This decreased mitochondrial density can be partly explained by reduced mitochondrial biogenesis as evidenced by the PGC1-α expression (Figure 4G) which only tended to be reduced (−16%, *p* = 0.085) in this group.

### 3.9. Inflammation

Systemic inflammation was evaluated by measuring the concentration of TNF-α in the plasma (Figure 5). Low in the sham-operated groups, the concentration of this inflammatory cytokine was strongly increased by sepsis in the DEF group (+205%), but not in the ALA group. In the EPA group, its value ranged between those measured in the DEF and ALA groups.

Cardiac inflammation was determined by evaluating the activation of the NFκB pathway and measuring the gene expression of IL-1β, IL-6 and TNF-α. Activation of the NFκB pathway was estimated by the decrease in IκBα protein expression. Except for the low value encountered in the sham-operated ALA-fed animals (−8 and −8% compared to the DEF and EPA groups), the other data were high and did not differ from one another. The activated NFκB pathway in the sham-operated animals of the ALA group did not increase the expression of TNF-α and IL-6. The expression of IL-1β was even reduced compared to that of the other two groups (−49 and 62% compared to the DEF and EPA groups).

However, sepsis increased the gene expression of IL-1β in all the dietary groups (+61, +189 and +102%, respectively) compared to the sham-operated animals of the DEF, ALA and EPA groups. For TNF-α mRNA, sepsis tended to induce similar effects, but the difference was significant only for the ALA and EPA groups (+169 and +103%, respectively). Finally, sepsis increased IL-6 mRNA only in the ALA group (+278% compared to the sham-operated group with the ALA substitution).

### 3.10. Cardiac Oxidative Stress in the Ex Vivo Situation

The levels of reduced and oxidized glutathione were appraised to measure the rate of reactive oxygen species (ROS) detoxification (Figure 6). If the amount of reduced glutathione was unaltered by the dietary manipulations and sepsis, those of oxidized glutathione depended on both of these factors. In the sham-operated animals, oxidized glutathione was higher in the DEF and ALA groups compared to the EPA group (+136 and +76%, respectively). Sepsis maintained a high level of oxidized glutathione in the DEF group, but strongly reduced it in the ALA group (−45%) and maintained it at a low level in the EPA group. Those changes reverberated logically on the GSH to GSSG ratio which is inversely proportional to the oxidative stress: it was low in the hearts of sham-operated animals fed with the DEF and ALA diets, as well as in those of septic rats fed with the DEF diet, but it was high in the other three subgroups; the sham-operated EPA group and the ALA and EPA sepsis groups.

The sepsis-induced decrease in oxidative stress in the ALA rats was apparently related to an increase in cardiac SOD2 content (+15% compared to the sham-operated animals of the same dietary group) without any change in SOD2 acetylation which is responsible for a reduced superoxide detoxification capacity. This was confirmed by a low SOD2 mRNA expression (−10% compared to the sham-operated animals of the same dietary group). These changes were not observed in the EPA group. Rather, the low oxidative stress observed in the septic EPA-fed rats was obviously due to an increased UCP3 content (+45% compared to the sham-operated animals of the same dietary group) deemed to reduce lipid-related ROS production. This last phenomenon was associated with a high SIRT3 mRNA content in the EPA-fed subgroups (+67% compared to the other two dietary groups).

## 4. Discussion

This study aimed at (i) verifying the cardioprotective effects of EPA in male rats made septic by CLP; (ii) evaluating the effects of ALA on the cardiac consequences of early sepsis; and (iii) determining probable physiological and molecular mechanisms responsible for the observed effects. Several characteristics indicate that the rats subjected to CLP were septic: (i) the concentrations of plasma triglycerides were significantly higher in the CLP groups compared to the sham-operated ones. Inflammation inhibits skeletal muscle β-oxidation in favor of glucose degradation, leading to accumulation of plasma triglycerides [27], (ii) the plasma TNF-α concentration was high in the DEF animals subjected to CLP, confirming the occurrence of a systemic inflammation; (iii) although the plasma concentrations of TNF-α were low in the CLP-subjected animals fed the ALA and EPA diets, the gene expressions of TNF-α and IL-1β in the heart were high, suggesting cardiac inflammation; (iv) the plasma ARA content was considerably reduced in the CLP groups. This progression is known to occur with sepsis and has already been demonstrated in humans [28]; (v) finally, the lean body mass was decreased in the septic groups, and this cachexic event is a hallmark of inflammation [29].

### 4.1. Ex Vivo Cardiac Function

The cardiac function was monitored in the ex vivo situation to get rid of all hormonal, nervous, systemic, but also bacterial, influences. It was determined in standardized conditions (composition of the perfusion fluid, temperature, left ventricular diastolic pressure) at constant flow (12 mL/min) with electrical pacing at 370 impulsions/min. Under these circumstances, cardiac mechanical activity reflects the health status of the cardiomyocytes and the perfusion pressure mirrors the dilatation status of the coronary microvessels.

In our study, we observed that the LVDP was reduced significantly by sepsis in the DEF group, whereas it tended to be increased in the other two groups. The sepsis-induced decrease in the LVDP of the DEF group was due to concomitant alterations of the contractility and relaxation, and it was strongly associated with a reduction of the coronary pressure which did not occur in the other 2 dietary groups. It is known that sepsis increases the expression of the COX-2 enzyme [3] which favors the synthesis of vasodilator agents (i.e., prostacyclin) from ARA. Since the DEF group contained the highest amount of ARA in the cardiac phospholipids, and the lowest content of COX2 inhibitor DHA, it is said that the highest prostacyclin production occurred in this group. Sepsis reduced thus the coronary pressure. However, the coronary flow was fixed to the value of 12 mL/min, and the oxygen and substrate supplies were the same for all the hearts whose weight did not differ between groups. It is thus unlikely that the reduced coronary pressure in this group was responsible for the decreased mechanical function. In contrast, it is more likely that the reduced mechanical activity was a consequence of mitochondrial damage. This hypothesis was strengthened by the sepsis-induced reduction of the myocardial oxygen consumption observed in the DEF group (−10%, *p* < 0.01). It was sustained by the increased lactate release to oxygen consumption ratio suggesting an increased part of anaerobic glycolysis in energy production and mitochondrial damage. It is asserted by the reduced VDAC content, which suggests a lower mitochondrial density partly due to a reduced expression of PGC1-α and to mitochondrial biogenesis. Finally, the hypothesis was also supported by the results of the cardiac GSH to GSSG ratio, which displayed a low value in the DEF diet-fed animals subjected to CLP. The high level of cellular ROS was probably responsible for an inhibition of some respiratory chain complexes [30]. In all likelihood, the ROS overproduction probably increased cellular damage, disrupted the mitochondria and reduced the cardiac mechanical function. In the other two dietary groups (ALA and EPA), the ex vivo cardiac function was not significantly altered and even tended to be increased. This clearly indicated that no cellular damage occurred in these last two groups. These conclusions are substantiated by the levels of circulating TNF-α which were high in the DEF diet-fed rats subjected to CLP and low in their homologues supplemented with n-3 PUFAs. Inflammatory cytokines, such as TNF-α and IL-1β, increase the calcium spike of the cardiomyocytes [31]. Therefore, the high circulating TNF-α level encountered in the septic DEF subgroup probably prompted a sharp increase in myocardial calcium to a high level and triggered mitochondrial dysfunctions and cardiomyocyte apoptosis. By contrast, the low level of TNF-α in the ALA and EPA groups reflected the anti-inflammatory effects of these PUFAs and their role in the preservation of cardiomyocyte health. ALA and EPA cardioprotective effects were related to a low oxidative stress: in the ALA group, this was associated with an increased mitochondrial SOD2 content and, thus, an increased capacity for superoxide detoxification. In the EPA group, this was associated with a high protein expression of UCP3, an uncoupling protein which reduces lipid-related ROS overproduction [32].

### 4.2. In Vivo Cardiac Function

When the cardiac function was determined in the in vivo situation, early sepsis led to a stimulation of the heart rate and contractility in the DEF and EPA groups, but not in the ALA group where the pathology triggered a significant decrease in the LVDP and contractility and a status quo for the heart rate. These data contrast with the results of the isolated hearts which displayed a strong sepsis-induced negative inotropic effect only in the DEF group. The major difference between the in vivo and ex vivo conditions was the coronary flow. In the ex vivo situation, the coronary flow was fixed and similar for all of the hearts. This was not the case in vivo, since the coronary flow depends on both the systolic aortic pressure and dilatation status of the coronary arterioles. After CLP, the lowest systolic aortic pressure was observed in the ALA group. On the other hand, the strongest coronary constriction occurred in the same group, as was indicated by the high ex vivo coronary pressure. Thus, it can be said that the in vivo coronary flow was the lowest in the ALA-CLP group. In a parallel study, we determined, ex vivo, the LVDP under conditions of increasing coronary flows, and we demonstrated a positive linear relation between the two factors: doubling the coronary flow multiplied the LVDP by a factor 2 (data not shown). Since we suspected that the in vivo coronary flow in the ALA group was the lowest among the groups of CLP-subjected animals, this would explain why the LVDP and contraction, in the ALA group, were also the most subdued. This could explain the low contractility observed, in vivo, in this group despite the fact that no cellular damage was noticed ex vivo. The in vivo coronary flow depends partly on the amount of PUFAs in the membrane phospholipids. PUFAs, mainly ARA and EPA, can be converted into vasodilator agents through the membrane-located COX activity. Since COX2 is induced during inflammation, we observed that the coronary pressure was reduced by sepsis. However, this was true for only the DEF and EPA groups. In the ALA group, this parameter tended to be increased. Analysis of the PUFA composition of the membrane phospholipids indicated that ARA and EPA were reduced in these animals and that the high proportion of DHA can inhibit COX2 [22]. In contrast, either ARA or EPA was high in the littermates fed the DEF and EPA diets. Sufficient amounts of vasodilation agents were apparently produced in these last 2 groups. This was not the case in the ALA group.

The high coronary flow expected in vivo in the DEF and EPA groups subjected to CLP enabled a good supply of oxygen to the myocardium and stimulated the heart rate. The lack of stimulation in the ALA group probably resulted from the suspected low in vivo coronary flow.

### 4.3. Efficiency of Oxygen Metabolism and Membrane PUFAs

In a previous study [19], we reported that feeding female rats with EPA improved the oxygen metabolism and was cardioprotective during sepsis. The phenomenon was characterized by an increased mitochondrial uncoupling protein 3 (UCP3) level which was responsible for lower mitochondrial oxidative stress. It was also associated with an increased mitochondrial SIRT3 content which is cardioprotective and allows the maintenance of energy metabolism. Our study suggests that the mechanism is probably similar in male rats.

In the present study, improved oxygen metabolism was also observed in the ALA group, but the cause was different: the oxidative stress was reduced due to the improvement in SOD2 activity (increased content with a maintained acetylation rate). The increased SOD2 protein expression probably resulted from a high IL-6 production as suggested by the high level of cardiac IL-6 mRNA. Indeed, high IL-6 has been shown to favor SOD2 expression in prostate, myeloma and brain cells [33,34,35] affording protection against interventions inducing a cellular oxidative stress. The high cardiac IL-6 expression probably helped to fight against oxidative stress by favoring SOD2 expression and ROS detoxification.

An improvement in oxygen metabolism was not observed with the DEF diet since the ratio of reduced to oxidized glutathione was extremely low in the heart after CLP, and this underlines the noxious action of excess TNF-α.

The improved oxygen metabolism in the EPA group was probably related to cellular EPA accumulation. Indeed, the increased SIRT3 mRNA expression as well as UCP3 and EPA contents occurred only in the EPA group. By contrast, the improved SOD2 activity after the ALA diet was not related to membrane EPA, since this PUFA was only present in trace amounts in this dietary group. The improved SOD2 activity can result from a high proportion of membrane docosapentaenoic acid (C22:5 n-3 or DPA) and/or a decreased long-chain n-6 PUFAs proportion (ARA, C22:4 n-6 and C22:5 n-6).

Finally, sepsis-induced vasodilation was partially triggered by ARA and EPA in both the DEF and EPA groups. Conversely, the sepsis-induced depression of vasodilation and contractile activity in the ALA group resulted from a deficiency in both ARA and EPA in the membrane phospholipids.

### 4.4. Inflammation, Oxidative Stress and Bactericidal Activity

Since the animals were fed diets with different PUFA compositions for several weeks before CLP induction, the fatty acid composition of lipopolysaccharides was most likely different when the pathology was triggered. This might result in altered toxicity against the organism. Our results, however, suggested that an inflammatory response was present in the three groups, although it was modulated by the dietary PUFAs. In the n-6 PUFA-rich group, a huge systemic TNF-α release was observed and explained myocardial toxicity. The release was not observed in the n-3 PUFA-enriched groups although the inflammation occurred through an increased TNF-α and IL-1β gene expression in the myocardium. By contrast, TNF-α and IL-1β mRNA expressions were not increased in the DEF group, probably because of a rapid mRNA translation. Thus, the fate of the inflammatory cytokine mRNA differed according to the characteristics of the membrane lipids. In the n-3 PUFA-rich heart, mRNA translation in proteins appeared slowed, perhaps aborted via the activity of DHA-related resolvins, protectins and/or maresins. This protected the heart against the noxious effects of CLP.

The low cytokine translation in the n-3 PUFA-rich hearts reduced systemic oxidative stress. It took different forms in the ALA and EPA groups: in the first one, a decreased lipid peroxidation was observed, whereas in the last one, an increased protein protection was noticed as suggested by the increased thiol content. These differential behaviors could be explained by the changes in the membrane lipid composition: the increased in vivo coronary flow suspected in the EPA group and the sepsis-induced stimulation of mechanical function suggest an activation of oxidative phosphorylation. In all likelihood, it resulted in a lower mitochondrial membrane potential which was associated with a decreased mitochondrial ROS release [36], protecting proteins against free-radical attack. The case of the ALA-rich hearts is less obvious. The diet did not increase the coronary flow and did not trigger sepsis-induced stimulation of the mechanical function. Rather, the cardiac mechanical function was depressed in vivo, which should reduce the energy demand, increase the mitochondrial membrane potential and increase the generation of ROS. The high ratio of reduced to oxidized glutathione observed after CLP, associated with increased SOD2 activity, suggested that the mitochondrial ROS detoxification was high and resulted in reduced lipid peroxidation. The phenomenon was also associated with a low plasma TNF-α level explaining the reduced oxidative stress.

TNF-α is a strong bactericidal agent [37]. It is produced during sepsis to kill the infectious bacteria. Of course, its toxic activity can turn against the host cells and trigger organ failure. In the DEF group, the high circulating TNF-α level probably prevented bacterial proliferation, but triggered cardiac failure. In the ALA and EPA groups, mRNA translation to TNF-α protein was partially or totally prevented. Taken together, all these data indicate that the sepsis-induced inflammatory response was attenuated by dietary ALA and EPA. The bactericidal capacities were thus low, and this may favor bacterial survival and/or proliferation. This could be detrimental for the organism in the long term.

### 4.5. Limitations of the Study

In the present study, we performed our measurements 24 h after sepsis induction to satisfy the requirements of the local committee of ethics for the animal experimentation and animal welfare structure of our laboratory in terms of animal comfort. Thus, we cannot certify that the long-term effects of EPA would be beneficial. However, data from the literature support this hypothesis. Indeed, numerous animal studies [15,16,17] report the long-term beneficial effects of dietary EPA-rich fish oils on the survival rate during sepsis.

Additionally, the parameters that we determined do not include measurements of apoptosis and fibrosis. It would have been interesting to obtain data describing these cellular phenomena to better characterize the beneficial effects of EPA. Similarly, the determination of the mechanisms of ALA would have been more complete with measurements of apoptosis and fibrosis since ex vivo cardiac function was high but in vivo contractility was low. Unfortunately, we did not have sufficient tissue to determine the different processes of cellular death in our study.

## 5. Conclusions

Our results show that LA-rich diets, which are characteristic of the Western societies, favor myocardial damage during sepsis. In contrast, EPA-fortified diets are cardioprotective in male rats, as has already been observed in female rats. This beneficial action is due to a double mechanism: (i) EPA has anti-inflammatory action, reduces the oxidative stress and preserves the energy metabolism through an increase in UCP3; (ii) incorporation of EPA in membrane phospholipids increases the vasodilator reserve of the coronary microvessels. Regarding diets enriched with ALA, the conclusion remains equivocal since the cardiac mechanical function was reduced in the in vivo situation. This last fatty acid displays anti-oxidative and protective effects through an increase in mitochondrial SOD2, but it does not allow the same increase in the vasodilator reserve as is observed with EPA. In conclusion, supplementing the Western diet with EPA can contribute to reducing sepsis-related mortality and to decreasing the health expenditures linked to this pathology. Our results underline the prime importance of dietary prevention in the management of sepsis. Dietary n-3 PUFAs seem to afford cardioprotection, but this effect is not sufficient to be beneficial for the heart in the in vivo situation. Membranes must be enriched with the vasodilator EPA to afford a real cardioprotection during sepsis. The association of EPA and ALA appears as a relevant strategy since it can increase the expression of UCP3 and SOD2, contributing thus to reducing ROS production and, at the same time, helping to increase the detoxification of superoxide radicals.

## Figures and Tables

**Figure 1 antioxidants-09-00371-f001:**
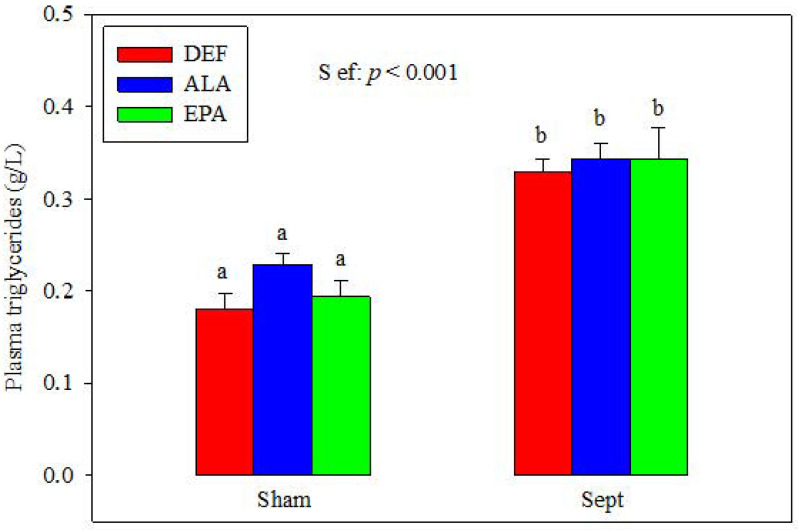
Plasma triglyceride concentrations. The means represent a consortium of 10 animals per group. DEF: n-3 polyunsaturated fatty acids (PUFA) deficient diet; ALA: diet enriched with α-linolenic acid; EPA: diet enriched with eicosapentaenoic acid; Sham: fictive operation; Sept: cecal ligation and puncture (CLP) performed to induce sepsis; S ef: sepsis effect; a, b: means without a common letter are significantly different.

**Figure 2 antioxidants-09-00371-f002:**
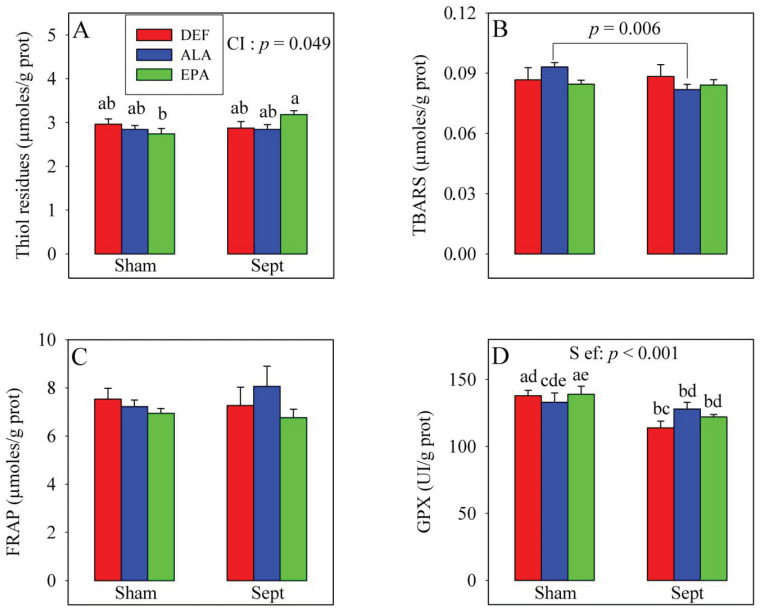
Parameters of the plasma oxidative stress. **A**: amounts of thiol residues; **B**: amounts of thiobarbituric acid reactive substances (TBARS); **C**: anti-oxidative defenses (FRAP); **D**: glutathione peroxidase (GPX) activities. The means represent a consortium of 10 animals per group. DEF: n-3 PUFA deficient diet; ALA: diet enriched with α-linolenic acid; EPA: diet enriched with eicosapentaenoic acid; Sham: fictive operation; Sept: operation performed to induce sepsis; TBARS: thiobarbituric acid reactive substances; FRAP: ferric reducing antioxidant power; GPX: glutathione peroxidase activity; S ef: sepsis effect; CI: cross interaction; a, b, c, d, e: means without a common letter in a same figure are significantly different.

**Figure 3 antioxidants-09-00371-f003:**
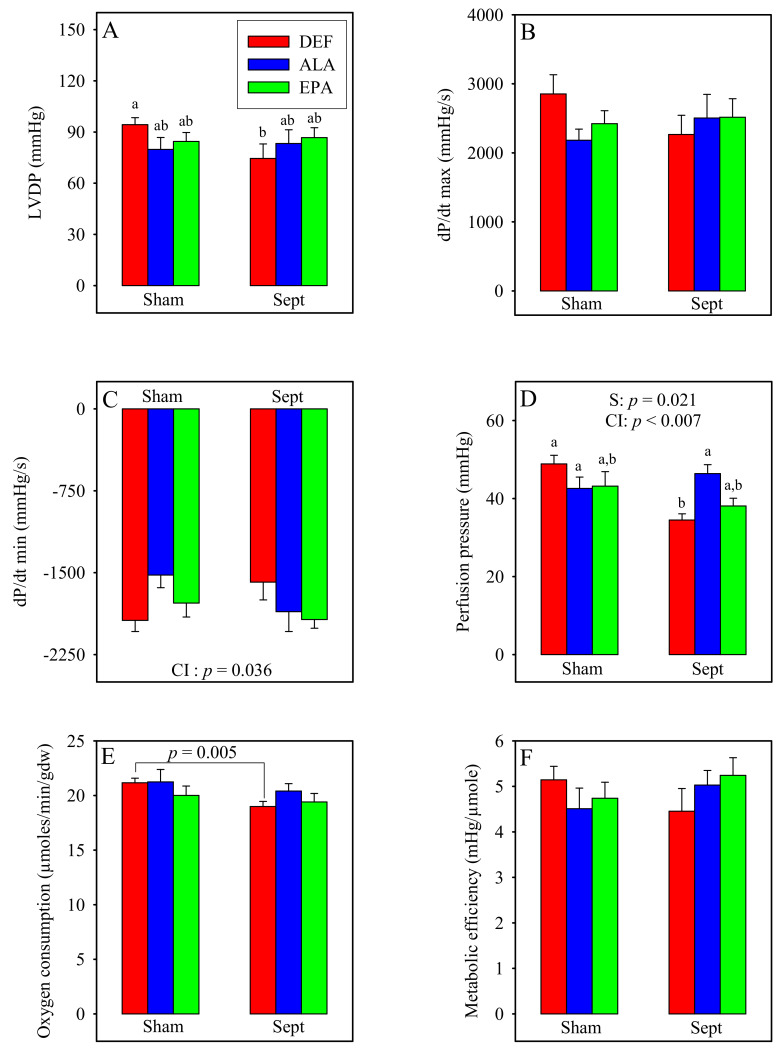
Activities of the isolated perfused hearts. **A**: left ventricular developed pressure; **B**: contraction; **C**: relaxation; **D**: perfusion pressure; **E**: oxygen consumption; **F**: metabolic efficiency. The means represent a consortium of 10 animals per group. DEF: n-3 PUFA deficient diet; ALA: diet enriched with α-linolenic acid; EPA: diet enriched with eicosapentaenoic acid; Sham: fictive operation; Sept: operation performed to induce sepsis; LVDP: left ventricular developed pressure; dP/dt max: contraction; dP/dt min: relaxation; S ef: sepsis effect; CI: cross interaction; a, b: means without a common letter in a same figure are significantly different.

**Figure 4 antioxidants-09-00371-f004:**
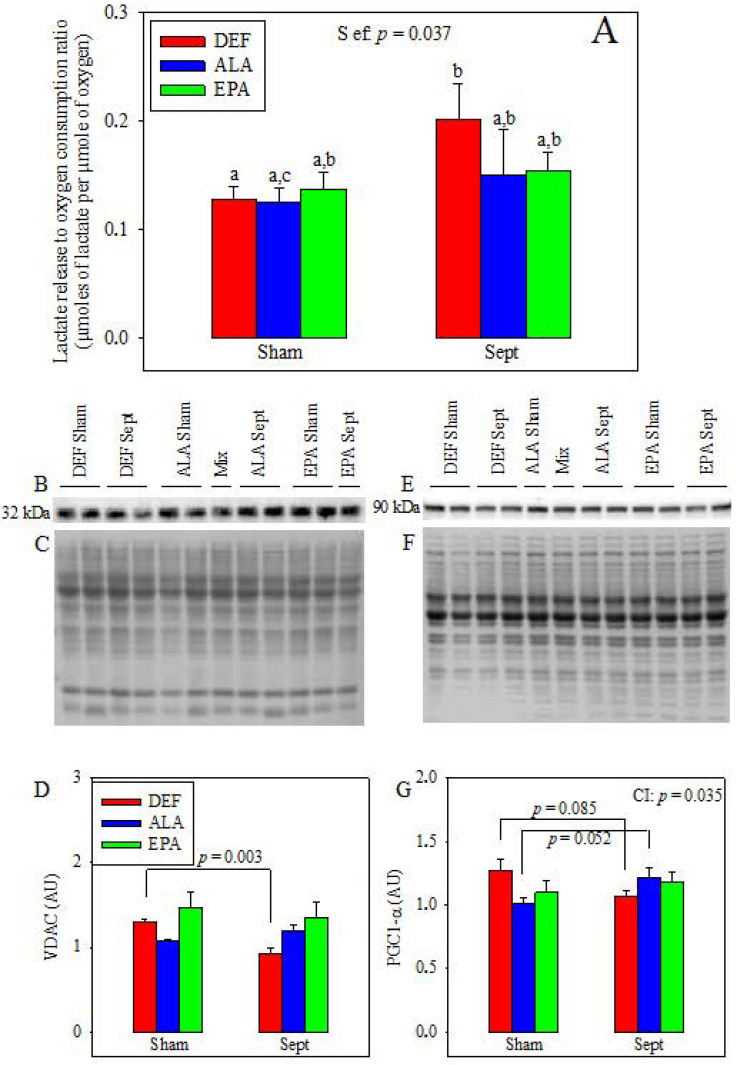
Appraisal of the mitochondrial function. (**A**) Lactate release in the coronary effluents of the isolated hearts to oxygen consumption ratio; (**B**) and (**E**) Voltage-dependent anion channel (VDAC) and peroxisome proliferator activated receptor gamma coactivator 1 alpha (PGC1-α) representative immunoblots; (**C**) and (**F**) Corresponding Ponceau red-colored total proteins; (**D**) and (**G**) Corresponding VDAC and PGC1-α expressions. The means represent a consortium of 10 animals per group. DEF: n-3 PUFA deficient diet; ALA: diet enriched with α-linolenic acid; EPA: diet enriched with eicosapentaenoic acid; Sham: fictive operation; Sept: operation performed to induce sepsis; a, b, c: means without a common letter in a same figure are significantly different.

**Figure 5 antioxidants-09-00371-f005:**
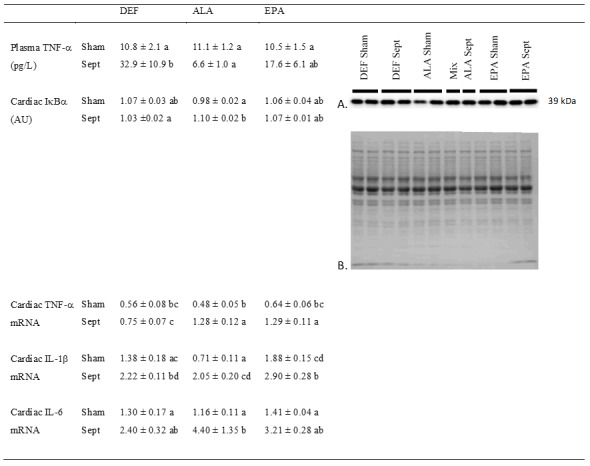
Inflammation markers. (**A**) Representative immunoblots of nuclear factor of kappa light polypeptide gene enhancer in B-cells inhibitor, alpha (IκBα); (**B**) Densitometry to red Ponceau S stain. The means represent a consortium of 10 animals per group. TNF-a: tumor necrosis factor a; IL1-β: interleukin 1-β; IL-6: interleukin 6; DEF: n-3 PUFA deficient diet; ALA: diet enriched with α-linolenic acid; EPA: diet enriched with eicosapentaenoic acid; Sham: fictive operation; Sept: operation performed to induce sepsis; a, b, c, d: means without a common letter in a same figure are significantly different.

**Figure 6 antioxidants-09-00371-f006:**
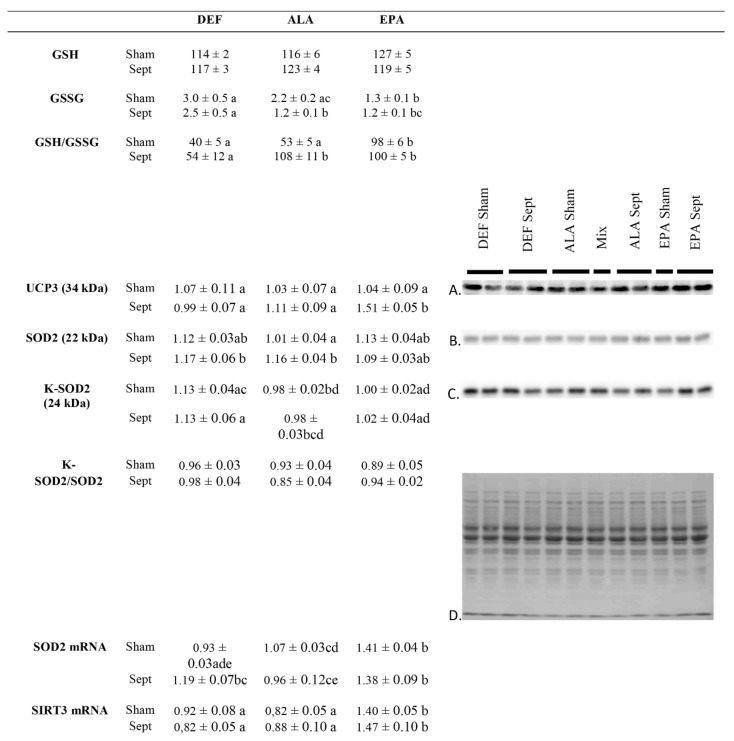
Parameters of cardiac oxidative stress. (**A**) Representative immunoblots of uncoupling protein 3 (UCP3); (**B**) Representative immunoblots of mitochondrial superoxide dismutase (SOD2); (**C**) Representative immunoblots of mitochondrial superoxide dismutase acetylated on lysine moieties (K-SOD2); (**D**) Densitometry to red Ponceau S stain. The means represent a consortium of 10 animals per group. GSH: reduced glutathione; GSSG: oxidized glutathione; SIRT3: sirtuin 3; DEF: n-3 PUFA deficient diet; ALA: diet enriched with α-linolenic acid; EPA: diet enriched with eicosapentaenoic acid; Sham: fictive operation; Sept: operation performed to induce sepsis; a, b, c, d, e: means without a common letter in a same figure are significantly different.

**Table 1 antioxidants-09-00371-t001:** Fatty acid composition of the 3 diets.

Fatty acid	DEF	ALA	EPA
C14:0	0.1	0.1	0.1
C16:0	6.5	6.5	5.7
C18:0	2.7	2.8	3.0
C20:0	0.1	0.1	0.1
C24:0	0.1	0.1	0.1
SFAs	9.5	9.6	9.0
C16:1 n-7	0.1	0,1	0.1
C18:1 n-9	40.8	30.0	26.6
C18:1 n-7	0.9	0.8	0.8
C20:1 n-9	0.1	0.1	0.4
MUFAs	42.0	31.1	27.9
C18:2 n-6	47.1	44.4	44.7
C20:2 n-6	Tr.	Tr.	0.2
C20:3 n-6	Tr.	Tr.	0.1
C20:4 n-6	Tr.	Tr.	0.8
n-6 PUFAs	48.1	44.4	46.9
C18:3 n-3	0.1	13.4	0.1
C20:4 n-3	Tr.	nd	0.4
C20:5 n3	nd	nd	15.5
n-3 PUFAs	0.1	13.4	15.6
n-6/n-3	481	3.3	3

DEF: n-3 PUFA deficient diet; ALA: diet enriched with α-linolenic acid; EPA: diet enriched with eicosapentaenoic acid; SFAs: saturated fatty acids; MUFAs: monounsaturated fatty acids; PUFAs: polyunsaturated fatty acids; n-6/n-3: ratio between n-6 and n-3 PUFAs; Tr.: trace amount; nd: not detected.

**Table 2 antioxidants-09-00371-t002:** Real-time Polymerase Chain Reaction (PCR) primers used in this study.

Gene	Sequence (5′-3′)
*β-Actin* (NM_031144.3)	(F) TCTGTGTGGATTGGTGGCTCTA(R) CTGCTTGCTGATCCACATCTG
*TNF-**α* (NM_012675.3)	(F) GCC TCT TCT CAT TCC TGC TC(R) GAG CCC ATT TGG GAA CTT CT
*IL-1**β* (NM_031512.2)	(F) AAATGCCTCGTGCTGTCTGA(R) GGTGTGCCGTCTTTCATCAC
*IL-6* (NM_012589.2)	(F) AGCGATGATGCACTGTCAGA(R) GGAACTCCAGAAGACCAGAGC
*SOD2* (NM_017051.2)	(F) TGAACAATCTGAACGTCACCG(R) CCTTAGGGCTCAGGTTTGTC
*SIRT3* (NM_001106313.2)	(F) TGCTACTCATTCTTGGGACCTC(R) CTGTACCGATTCAGACAAGCTG

(F): forward; (R): reverse; TNF-α: tumor necrosis factor alpha; IL-1β: interleukin-1β; IL-6: interleukin-6; SOD2: superoxide dismutase 2; SIRT3: sirtuin 3.

**Table 3 antioxidants-09-00371-t003:** Morphological data.

Tissue	DEF Sham	DEF Sept	ALA Sham	ALA Sept	EPA Fict	EPA Sept	*p*-Value
Body weight	552 ± 22	549 ± 15	581 ± 24	565 ± 18	590 ± 22	578 ± 20	
Δ body weight	−3.3 ± 0.6	−2.0 ± 0.8	−3.0 ± 0.4	−2.5 ± 0.3	−2.1 ± 0.3	−2.2 ± 0.3	
Fat mass	12.5 ± 1.1	12.4 ± 0.9	14.6 ± 0.8	15.5 ± 0.9	15.9 ± 1.1	14.6 ± 1.2	D ef: *p* = 0.021
Δ fat mass	−0.34 ± 0.19 a, b	−0.17 ± 0.31 b	−0.87 ± 0.15 a, c	−0.01 ± 0.12 b	−0.97 ± 0.08 a	−0.39 ± 0.17 b, c	S ef: *p* = 0.001
Lean mass	79.0 ± 1.1 a	79.6 ± 0.9 a, b	77.5 ± 0.9 a, b	76.4 ± 1.0 b	75.7 ± 1.0 a, b	76.9 ± 1.3 a, b	D ef: *p* = 0.032
Δ lean mass	−2.61 ± 0.50 a, b	−3.10 ± 0.68 b, c	−2.35 ± 0.32 a, c, d, e	−3.59 ± 0.26 b	−1.33 ± 0.37 a, d	−2.66 ± 0.36 b, e	D ef: *p* = 0.025 S ef: *p* = 0.003
Water	57.3 ± 1.0 a, b	57.1 ± 0.6 a	55.5 ± 0.6 a, b	55.0 ± 0.8 b	54.3 ± 0.9 a, b	55.0 ± 0.9 a, b	D ef: *p* = 0.014
Δ water	−1.89 ± 0.63 a, b	−0.71 ± 0.49 a, b	−1.15 ± 0.29 a, b	−2.05 ± 0.37 a	−0.39 ± 0.37 b	−0.34 ± 0.38 b	D ef: *p* = 0.020
PAT	1.28 ± 0.11 a, b	1.25 ± 0.10 a	1.31 ± 0.07 a	1.61 ± 0.10 b	1.65 ± 0.08 ab	1.49 ± 0.10 ab	D ef: *p* = 0.024
VAT	1.43 ± 0.05 a	1.64 ± 0.10 a, c	1.68 ± 0.10 a, c	1.85 ± 0.13 b, c	1.86 ± 0.10 b, c	2.09 ± 0.15 b	D ef: *p* < 0.001 S ef: *p* = 0.048
AAT	2.77 ± 0.17 a	2.89 ± 0.19 a, b	2.99 ± 0.15 a, b	3.46 ± 0.19 a, b	3.60 ± 0.11 b	3.49 ± 0.27 a, b	D ef: *p* = 0.002
EAT	1.25 ± 0.12	1.29 ± 0.08	1.33 ± 0.09	1.48 ± 0.08	1.36 ± 0.07	1. 44 ± 0.07	

The data represent the mean and standard error of 10 animals per group. The body weight is expressed in g, the weight of the different adipose tissues, as well as the variations in weight, are expressed in g/100 g of body weight. DEF Sham: sham-operated animals fed with the n-3 PUFA deficient diet; DEF Sept: septic animals fed with the n-3 PUFA deficient diet; ALA Sham: sham-operated animals fed with the ALA-rich diet; ALA Sept: septic animals fed with the ALA-rich diet; EPA Sham: sham-operated animals fed with the EPA-rich diet; EPA Sept: septic animals fed with the EPA-rich diet; Δ: variation; PAT: perirenal adipose tissue; VAT: visceral adipose tissue; AAT: abdominal adipose tissue (sum of peri-renal and visceral adipose tissues); EAT: epididymal adipose tissue; p-value: probability value; D ef: diet effects; S ef: sepsis effects; a, b, c, d, e: the data on a same line without a common letter are significantly different.

**Table 4 antioxidants-09-00371-t004:** Fatty acid composition of plasma lipids.

Fatty acid	DEF Sham	DEF Sept	ALA Sham	ALA Sept	EPA Sham	EPA Sept	*p*-Value
C14:0	0.41 ± 0.04	1.22 ± 0.77	0.39 ± 0.03	0.52 ± 0.10	0.38 ± 0.04	0.45 ± 0.04	
C15:0	0.16 ± 0.01	0.14 ± 0.01	0.16 ± 0.01	0.16 ± 0.02	0.14 ± 0.02	0.15 ± 0.01	S ef: *p* < 0.001
C16:0	16.53 ± 0.56 a	18.59 ± 0.48 b	17.05 ± 0.32 a, c	17.79 ± 0.48 b, c, d	16.99 ± 0.39 a, d	18.43 ± 0.47 b, c	S ef: *p* < 0.05
C17:0	0.23 ± 0.01 a	0.21 ± 0.01 a, b	0.21 ± 0.01 a, b	0.20 ± 0.01 a, b	0.22 ± 0.01 a, b	0.20 ± 0.01 b	S ef: *p* = 0.035
C18:0	13.57 ± 0.59	13.95 ± 0.65	13.84 ± 0.43	14.96 ± 0.62	12.44 ± 1.42	14.39 ± 0.44	
C20:0	0.08 ± 0.03	0.14 ± 0.03	0.08 ± 0.02	0.12 ± 0.02	0.12 ± 0.03	0.15 ± 0.03	
C22:0	0.20 ± 0.02	0.21 ± 0.02	0.22 ± 0.01	0.24 ± 0.01	0.24 ± 0.01	0.22 ± 0.02	
C24:0	0.49 ± 0.02 a, b	0.50 ± 0.01 a, b	0.51 ± 0.02 a, b	0.50 ± 0.02 a, b	0.53 ± 0.02 a	0.47 ± 0.02 b	
DMA 16:0	0.14 ± 0.01 a, b	0.11 ± 0.01 a, b	0.13 ± 0.02 a, b	0.10 ± 0.01 a	0.13 ± 0.02 b	0.13 ± 0.01 ab	
DMA 18:0	0.07 ± 0.02 a, c	0.01 ± 0.01 b, d	0.06 ± 0.02 a, d, e	0.01 ± 0.01 b, c, e	0.09 ± 0.02 a	0.02 ± 0.01 b, c, e	S ef: *p* < 0.001
SFA	32.10 ± 0.45 a	35.3 ± 1.21 b, c	32.89 ± 0.37 a	34.86 ± 0.51 b	31.51 ± 1.29 a, c	34.83 ± 0.28 b	S ef: *p* < 0.001
C16:1 n-7	1.20 ± 0.18 a	1.19 ± 0.06 a, c	1.29 ± 0.12 a, c	1.44 ± 0.12 b, c	1.07 ± 0.01 a	1.55 ± 0.17 b, c	D ef: *p* = 0.048 S ef: *p* < 0.001
C18:1 n-7	2.10 ± 0.09 a	2.06 ± 0.10 a	2.06 ± 0.08 a	2.09 ± 0.10 a	1.88 ± 0.10 b	1.93 ± 0.07 a	D ef: *p* = 0.003
C18:1 n-9	12.60 ± 1.03 a, c	14.67 ± 0.86 b	10.85 ± 0.43 a	12.86 ± 0.80 c	10.08 ± 0.52 a	12.62 ± 0.60 b, c	D ef: *p* = 0.009 S ef: *p* < 0.001
C20:1 n-9	0.15 ± 0.01 a, b	0.17 ± 0.01 a	0.13 ± 0.01 b, c	0.16 ± 0.01 a, d	0.15 ± 0.01 b, d, e	0.15 ± 0.01 a, c, e	S ef: *p* = 0.004
C22:1 n-9	0.06 ± 0.02 a, b	0.09 ± 0.02 b, c	0.03 ± 0.02 a	0.09 ± 0.02 b	0.05 ± 0.02 a, c	0.11 ± 0.01 b	S ef: *p* < 0.001
C24:1 n-9	0.36 ± 0.02 a	0.42 ± 0.02 a, b	0.41 ± 0.02 a, b	0.46 ± 0.02 b	0.46 ± 0.02 b	0.44 ± 0.03 a, b	D ef: *p* = 0.040 CI: *p* = 0.035
MUFA	16.66 ± 1.24 a, c	18.76 ± 0.95 b	14.94 ± 0.47 a, d	17.23 ± 0.99 b, c, d	15.29 ± 1.83 a	16.94 ± 0.76 b, c, d	S ef: *p* < 0.001
C18:2 n-6	11.96 ± 0.92 a	13.11 ± 0.77 b, d, e	13.05 ± 0.55 a, e	15.20 ± 0.74 b, f	14.06 ± 0.53 b, e	15.53 ± 0.39 c, d, f	D ef: *p* < 0.001 S ef: *p* < 0.001
C18:3 n-6	0.25 ± 0.02 a	0.14 ± 0.01 b	0.22 ± 0.01 a	0.14 ± 0.01 b	0.15 ± 0.02 b	0.12 ± 0.02 b	D ef: *p* = 0.002 S ef: *p* < 0.001 CI: *p* = 0.015
C20:2 n-6	0.17 ± 0.01 a, c	0.18 ± 0.01 a, c	0.19 ± 0.01 a, c	0.21 ± 0.01 b	0.20 ± 0.01 b, c	0.19 ± 0.01 b, c	D ef: *p* = 0.047
C20:3 n-6	0.28 ± 0.02 a	0.21 ± 0.02 a	0.44 ± 0.03 b, c	0.42 ± 0.04 b	0.52 ± 0.03 c	0.47 ± 0.04 b, c	D ef: *p* < 0.001 S ef: *p* = 0.048
C20:4 n-6	34.34 ± 1.90 a	27.62 ± 0.74 d, e	31.81 ± 0.97 b	25.71 ± 1.26 d	28.48 ± 1.04 e	22.16 ± 0.86 c	D ef: *p* < 0.001 S ef: *p* < 0.001
C22:4 n-6	0.74 ± 0.02 a	0.81 ± 0.06 a	0.32 ± 0.02 b	0.30 ± 0.01 b	0.23 ± 0.02 b, d	0.20 ± 0.02 c, d	D ef: *p* < 0.001
C22:5 n-6	2.58 ± 0.22 a	2.85 ± 0.15 a	0.22 ± 0.01 b	0.27 ± 0.02 b	0.13 ± 0.02 b	0.15 ± 0.01 b	D ef: *p* < 0.001
n-6 PUFA	50.32 ± 1.29 a	44.91 ± 0.85 c, d	46.24 ± 0.62 c	42.25 ± 0.97 d	43.78 ± 0.64 c, d	38.83 ± 0.71 b	D ef: *p* < 0.001 S ef: *p* < 0.001
C18:3 n-3	nd a	nd a	0.51 ± 0.04 b	0.77 ± 0.09 c	0.02 ± 0.02 a	0.01 ± 0.01 a	D ef: *p* < 0.001 S ef: *p* = 0.011 CI: *p* = 0.001
C20:5 n-3	0.07 ± 0.02 a	0.09 ± 0.02 a	0.73 ± 0.06 b	0.63 ± 0.04 b, a	3.78 ± 0.32 c	3.75 ± 0.26 c	D ef: *p* < 0.001
C22:5 n-3	0.08 ± 0.02 a	0.10 ± 0.02 a	0.53 ± 0.02 b	0.48 ± 0.02 b	1.19 ± 0.08 c	1.33 ± 0.05 d	D ef: *p* < 0.001 S ef: *p* = 0.007 CI: *p* < 0.001
C22:6 n-3	0.95 ± 0.03 a	0.99 ± 0.03 a	4.38 ± 0.12 b	3.99 ± 0.16 c	4.60 ± 0.12 b	4.51 ± 0.15 b	D ef: *p* < 0.001
n-3 PUFA	1.10 ± 0.04 a	1.20 ± 0.04 a	6.15 ± 0.13 b	5.87 ± 0.11 b	9.63 ± 0.30 c	9.60 ± 0.29 c	D ef: *p* < 0.001
PUFA	51.42 ± 1.29 a, d, f	46.10 ± 0.83 b, e	52.39 ± 0.56 c, f	48.11 ± 0.98 b, d, e	53.41 ± 0.67 c, f	48.43 ± 0.88 b, d	S ef: *p* < 0.001
n-6/n-3	46.3 ± 2.16 a	38.1 ± 1.8 b	7.6 ± 0.2 c	7.2 ± 0.2 c	4.6 ± 0.1 c	4.1 ± 0.1 c	D ef: *p* < 0.001 S ef: *p* = 0.002 CI: *p* = 0.002

The results include 10 animals per group. DEF Sham: sham-operated animals fed with the n-3 PUFA deficient diet; DEF Sept: septic animals fed with the n-3 PUFA deficient diet; ALA Sham: sham-operated animals fed with the ALA-rich diet; ALA Sept: septic animals fed with the ALA-rich diet; EPA Sham: sham-operated animals fed with the EPA-rich diet; EPA Sept: septic animals fed with the EPA-rich diet; DMA: dimethyl acetal; SFA: saturated fatty acids; MUFA: monounsaturated fatty acids; PUFA: polyunsaturated fatty acids; n-6/n-3: ratio between n-6 and n-3 PUFA; nd: not detected; D ef: diet effects; S ef: sepsis effects; CI: cross-interaction. a, b, c, d, e, f: the data printed on a same line without a common letter are significantly different.

**Table 5 antioxidants-09-00371-t005:** In vivo cardiac function.

Parameter	DEF Sham	DEF Sept	ALA Sham	ALA Sept	EPA Sham	EPA Sept	*p*-Value
	Aorta						
ASP	88.3 ± 6.5	86.0 ± 3.1	91.9 ± 5.2	84.3 ± 4.0	91.1 ± 6.1	82.8 ± 3.8	
ADP	63.8 ± 5.4	58.6 ± 3.7	58.7 ± 3.5	55.1 ± 4.1	64.7 ± 6.5	52.6 ± 6.5	
AMP	76.1 ± 5.9	72.3 ± 3.4	72.2 ± 3.6	69.7 ± 3.9	77.9 ± 6.3	67.7 ± 5.1	
	Heart						
LVDP	93 ± 2 ab	99 ± 3 ab	107 ± 5 a	87 ± 4 b	103 ± 6 ac	102 ± 3 ab	CI: *p* = 0.032
HR	246 ± 7 ab	271 ± 11 ab	259 ± 8 ab	257 ± 16 ab	233 ± 7 a	289 ± 10 b	S ef: *p* = 0.004 CI: *p* = 0.027
dP/dt max	5183 ± 189 ab	5504 ± 238 ab	5579 ± 355 a	4244 ± 569 b	5034 ± 410 ab	5533 ± 341 ab	CI: *p* < 0.045
dP/dt min	−4549 ± 455	−4364 ± 204	−4294 ± 361	−3416 ± 349	−4244 ± 463	−4664 ± 130	

The data represent the means and SEM of 10 samples per group. DEF Sham: sham-operated animals fed with the n-3 PUFA deficient diet; DEF Sept: septic animals fed with the n-3 PUFA deficient diet; ALA Sham: sham-operated animals fed with the ALA-rich diet; ALA Sept: septic animals fed with the ALA-rich diet; EPA Sham: sham-operated animals fed with the EPA-rich diet; EPA Sept: septic animals fed with the EPA-rich diet; ASP: aortic systolic pressure; ADP: aortic diastolic pressure; AMP: aortic mean pressure; LVDP: left ventricular developed pressure; HR: heart rate; dp/dt max: contraction; dP/dt min: relaxation. Systolic and diastolic pressures, heart rate and contraction/relaxation are expressed in mmHg, beats/min and mmHg/s. vp-value: probability value; S ef: Sepsis effects; CI: cross interaction between the effects of the diets and those of sepsis. a, b, c: the data on a same line without a common letter are significantly different.

**Table 6 antioxidants-09-00371-t006:** Fatty acid composition of cardiac phospholipids.

Fatty acid	DEF Sham	DEF Sept	ALA Sham	ALA Sept	EPA Sham	EPA Sept	*p*-Value
C16:0	10.76 ± 0.38	10.69 ± 0.38	10.71 ± 0.18	10.59 ± 0.15	11.18 ± 0.30	11.02 ± 0.24	
C18:0	22.04 ± 0.11 a	21.87 ± 0.16 a, c	21.28 ± 0.18 b, d	21.28 ± 0.24 c, d	20.70 ± 0.17 b	21.27 ± 0.18 c, d	D ef: *p* < 0.001
DMA C16:0	2.03 ± 0.08 a	2.17 ± 0.08 a, c	2.53 ± 0.23 b	2.51 ± 0.13 b	2.41 ± 0.06 b, c	2.41 ± 0.06 b, c	D ef: *p* < 0.001
DMA C18:0	0.89 ± 0.09	0.88 ± 0.04	1.02 ± 0.04	0.87 ± 0.12	0.96 ± 0.04	0.96 ± 0.05	
C20:0	0.11 ± 0.01	0.08 ± 0.02	0.09 ± 0.02	0.10 ± 0.03	0.09 ± 0.02	0.07 ± 0.03	
C22:0	0.15 ± 0.04	0.13 ± 0.05	0.17 ± 0.02	0.19 ± 0.03	0.18 ± 0.02	0.15 ± 0.04	
SFA	36.37 ± 0.34	36.16 ± 0.41	36.11 ± 0.32	35.90 ± 0.20	35.86 ± 0.28	36.23 ± 0.28	
C16:1 n-9	0.13 ± 0.03 a	0.06 ± 0.02 a, c	nd b, d	0.03 ± 0.03 b, d	0.04 ± 0.02 a, d, e	0.03 ± 0.03 b, c, e	D ef: *p* < 0.001
C16:1 n-7	0.24 ± 0.02 a	0.35 ± 0.05 b	0.29 ± 0.02 a, b	0.33 ± 0.03 a, b	0.32 ± 0.05 a, b	0.290 ± 0.03 a, b	
C18:1 n-9	5.71 ± 0.22 a	5.29 ± 0.19 b	4.54 ± 0.07 c	4.73 ± 0.23 c	4.65 ± 0.12 c	4.86 ± 0.13 bc	D ef: *p* < 0.001 CI: *p* = 0.023
C18:1 n-7	3.77 ± 0.09 a, c, e	3.96 ± 0.06 a	3.75 ± 0.10 a, c, e	3.94 ± 0.08 a, d	3.67 ± 0.07 b, d, e	3.63 ± 0.17 b, c	
C22:1 n-9	0.16 ± 0.07	0.06 ± 0.02	nd	0.04 ± 0.04	0.06 ± 0.06	0.06 ± 0.06	
MUFA	10.09 ± 0.26 a	9.79 ± 0.23 a, c	8.69 ± 0.15 b, d	9.18 ± 0.26 a, d, e	8.85 ± 0.25 b, e	8.91 ± 0.26 b, c, e	D ef: *p* < 0.001
C18:2 n-6	14.82 ± 0.46 a	14.38 ± 0.84 a	17.12 ± 0.77 b, f, g, h	17.86 ± 0.81 c, g, i	18.67 ± 0.59 d, f	17.17 ± 0.35 e, h, i	D ef: *p* < 0.001 S ef: *p* = 0.004
C20:2 n-6	0.11 ± 0.03	0.12 ± 0.01	0.13 ± 0.01	0.13 ± 0.01	0.14 ± 0.01	0.16 ± 0.01	
C20:3 n-6	0.19 ± 0.02 a	0.23 ± 0.02 a	0.30 ± 0.02 b	0.30 ± 0.01 b	0.33 ± 0.02 b	0.33 ± 0.01 b	D ef: *p* < 0.001
C20:4 n-6	26.99 ± 0.27 a	25.72 ± 0.49 b	23.30 ± 0.53 e	23.43 ± 0.57 e	19.88 ± 0.50 c	21.35 ± 0.46 d	D ef: *p* < 0.001 CI: *p* = 0.010
C22:4 n-6	2.17 ± 0.11 a	2.30 ± 0.08 a	0.87 ± 0.03 b	0.84 ± 0.03 b	0.40 ± 0.04 c	0.54 ± 0.01 c	D ef: *p* < 0.001
C22:5 n-6	7.74 ± 0.17 a	9.36 ± 0.52 b	0.62 ± 0.04 c	0.55 ± 0.07 c	0.21 ± 0.03 c	0.26 ± 0.03 c	D ef: *p* < 0.001 S ef: *p* = 0.012 CI: *p* = 0.001
n-6 PUFA	52.03 ± 0.40 a	52.12 ± 0.35 a	42.35 ± 0.38 b	43.11 ± 0.46 b	39.63 ± 0.48 c	39.81 ± 0.51 c	D ef: *p* < 0.001
C18:3 n-3	nd a	nd a	0.22 ± 0.01 b	0.22 ± 0.01 b	nd a	nd a	D ef: *p* < 0.001
C20:5 n-3	nd a	nd a	nd a	0.02 ± 0.02 a	0.79 ± 0.07 b	0.66 ± 0.02 c	D ef: *p* < 0.001
C22:5 n-3	0.12 ± 0.01 a	0.17 ± 0.01 a	1.93 ± 0.13 b	1.84 ± 0.13 b	4.17 ± 0.24 c	4.07 ± 0.16 c	D ef: *p* < 0.001
C22:6 n-3	1.46 ± 0.13 a	1.73 ± 0.07 a	10.62 ± 0.35 b, d	9.70 ± 0.36 c, e	10.61 ± 0.35 b, d, e	10.24 ± 0.25 b, d, e	D ef: *p* < 0.001
n-3 PUFA	1.52 ± 0.12 a	1.93 ± 0.07 a	12.85 ± 0.34 b	11.81 ± 0.36 c	15.66 ± 0.31 d	15.06 ± 0.33 e	D ef: *p* < 0.001 S ef: *p* = 0.048 CI: *p* = 0.019
n-6/n-3	35.2 ± 3.1 a	27.2 ± 1.0 b	3.3 ± 0.1 c	3.7 ± 0.1 c	2.5 ± 0.1 c	2.7 ± 0.09 c	D ef: *p* < 0.001 S ef: *p* = 0.045 CI: *p* = 0.009
Total PUFA	53.54 ± 0.28 a	54.05 ± 0.40 a, c	55.20 ± 0.22 b, d	54.92 ± 0.32 c, d, e	55.28 ± 0.23 b, e, f	54.87 ± 0.21 c, d, f	D ef: *p* < 0.001

The data represent the means and standard errors of the mean of 5 samples, randomly chosen in each group. DEF Sham: sham-operated animals fed with the n-3 PUFA deficient diet; DEF Sept: septic animals fed with the n-3 PUFA deficient diet; ALA Sham: sham-operated animals fed with the ALA-rich diet; ALA Sept: septic animals fed with the ALA-rich diet; EPA Sham: sham-operated animals fed with the EPA-rich diet; EPA Sept: septic animals fed with the EPA-rich diet; DMA: dimethyl acetal; SFA: saturated fatty acids; MUFA: monounsaturated fatty acids; PUFA: polyunsaturated fatty acids; n-6/n-3: ratio between n-6 and n-3 PUFA; nd: not detected; p-value: probability value; D ef: diet effects; S ef: sepsis effects; CI: cross-interaction. a, b, c, d, e, f, g, h, i: the data printed on a same line without a common letter are significantly different.

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
