# Peer review of "EPA Is Cardioprotective in Male Rats Subjected to Sepsis, but ALA is Not Beneficial"

_antioxidants, 2020, doi:10.3390/antiox9050371_

Round 1
Reviewer 1 Report
In the study titled, "EPA is cardioprotective in male rats subjected to sepsis, but ALA is not beneficial", Leger T et al., document the impact of EPA and ALA on cardiac function in a sepsis model of injury. Authors show that EPA administration in diet has a cardioprotective effect via modulation of inflammatory processes and reducing oxidative stress. Overall, the hypothesis is interesting and the results are well organized. There are few comments;
- The mechanism is less clear and the ability of EPA or ALA to target various signaling processes being shown in the results is not well tested.
- It is unclear whether cardiac parameters ex vivo and in vivo were recorded within 24 hours of sepsis. If that is the case, the long term effect on cardiac function and whether the decrease in function after 24 hours holds remains unclear.
- The title states cardioprotective effect, however there in data on cardiomyocyte apoptosis or changes in necrosis in the heart represents a limitation.
Author Response
My colleagues and I would like to thank the reviewer for his constructive remarks. Here are our responses to his comments:
- The mechanism is less clear and the ability of EPA or ALA to target various signaling processes being shown in the results is not well tested.
To clarify the mechanism of ALA and EPA, the known influence of each fatty acid on the lipid composition of cardiac phospholipids has been presented in the Introduction section. We have also described the potential action of these two fatty acids on the production of vasoactive compounds, coronary flow and myocardial contractility. Finally, 2 sentences have been added to the conclusion of the manuscript to improve the understanding of the mechanisms of EPA and ALA.
- It is unclear whether cardiac parameters ex vivo and in vivo were recorded within 24 hours of sepsis. If that is the case, the long term effect on cardiac function and whether the decrease in function after 24 hours holds remains unclear.
We detailed the fact that the measurements of the cardiac function were performed 24 h after sepsis induction in the Materials and Methods section. A paragraph describing the limitations of our study was added in the Discussion section. In this paragraph, the fact that our measurements was performed 24 h after sepsis induction was presented as a limitation of our study and it was discussed according to the literature data.
- The title states cardioprotective effect, however there in data on cardiomyocyte apoptosis or changes in necrosis in the heart represents a limitation.
Concerning the measurements of apoptosis and fibrosis, we added a paragraph in the Limitations of our study.
PS: all the corrections done are in red ink in the text.
Reviewer 2 Report
In my opinion, manuscript entitled ,,EPA IS CARDIOPROTECTIVE IN MALE RATS SUBJECTED TO SEPSIS, BUT ALA IS NOT BENEFICIAL” was carefully written, and the experiment itself was carefully planned. I have, however, reservations regarding the arrangement of animals and the statistical methods used.
I suppose the cage distribution was as follows: 333344 in each dietary group, then it was divided into 334 (Sept and Sham). In this situation, the statistical unit should be a cage, not an animal, so n=3, not n=10. Even if there are analogous publications in which the statistical unit is an animal and if you have so calculated it so far, this is a mistake. Of course, n=3 are not enough to make reliable statistical comparisons, so it can't be changed now, but keep the above remark in mind in the future experiment planning.
My second remark concerns the statistical test used (LSD). This is not a good comparison test, you should use the Tukey HSD test or Duncan test.
I would suggest that in the tables the last column indicates the exact p value from the general comparison of the means, e.g. 0.042, 0.007, <0.001. Then the column name could be ,,p-value”.
Author Response
My colleagues and I would like to thank the reviewer for his constructive remarks. Here are our responses to his comments:
- I suppose the cage distribution was as follows: 333344 in each dietary group, then it was divided into 334 (Sept and Sham). In this situation, the statistical unit should be a cage, not an animal, so n=3, not n=10. Even if there are analogous publications in which the statistical unit is an animal and if you have so calculated it so far, this is a mistake. Of course, n=3 are not enough to make reliable statistical comparisons, so it can't be changed now, but keep the above remark in mind in the future experiment planning.
We appreciate the scientific discipline of the reviewer. However, as indicat;ed in the description of the surgical procedure, the animals were placed in individual cages from the end of surgery. Each rat thus followed an individual destiny related to the effect of dietary PUFAs and the surgical procedure.
- My second remark concerns the statistical test used (LSD). This is not a good comparison test, you should use the Tukey HSD test or Duncan test.
We apologize for the utilization of the LSD test as a post-hoc test. We recalculated all the statistical data by using the Duncan’s test and corrected figures and tables with the new data.
- I would suggest that in the tables the last column indicates the exact p value from the general comparison of the means, e.g. 0.042, 0.007, <0.001. Then the column name could be ,,p-value”.
The exact p-values were given in the tables and figures.
PS: all the corrections done are in red ink in the text.
Round 2
Reviewer 1 Report
No further comments